# Studies on the Exposure of Gadolinium Containing Nanoparticles with Monochromatic X-rays Drive Advances in Radiation Therapy

**DOI:** 10.3390/nano10071341

**Published:** 2020-07-09

**Authors:** Fuyuhiko Tamanoi, Kotaro Matsumoto, Tan Le Hoang Doan, Ayumi Shiro, Hiroyuki Saitoh

**Affiliations:** 1Institute for Integrated Cell-Material Sciences, Institute for Advanced Study, Kyoto University, Kyoto 606-8501, Japan; matsumoto.kotaro.5r@kyoto-u.ac.jp; 2Department of Microbio., Immunol. & Molec. Genet., University of California, Los Angeles, CA 90095, USA; 3Center for Innovative Materials and Architectures (INOMAR), Vietnam National University-Ho Chi Minh City, Ho Chi Minh City 721337, Vietnam; dlhtan@inomar.edu.vn; 4Kansai Photon Science Institute, Quantum Beam Science Research Directorate, National Institutes for Quantum and Radiological Science and Technology, Hyogo 679-0198, Japan; shiro.ayumi@qst.go.jp (A.S.); saito.hiroyuki@qst.go.jp (H.S.)

**Keywords:** monochromatic X-ray, mesoporous silica nanoparticles, high-Z elements, tumor spheroids

## Abstract

While conventional radiation therapy uses white X-rays that consist of a mixture of X-ray waves with various energy levels, a monochromatic X-ray (monoenergetic X-ray) has a single energy level. Irradiation of high-Z elements such as gold, silver or gadolinium with a synchrotron-generated monochromatic X-rays with the energy at or higher than their K-edge energy causes a photoelectric effect that includes release of the Auger electrons that induce DNA damage—leading to cell killing. Delivery of high-Z elements into cancer cells and tumor mass can be facilitated by the use of nanoparticles. Various types of nanoparticles containing high-Z elements have been developed. A recent addition to this growing list of nanoparticles is mesoporous silica-based nanoparticles (MSNs) containing gadolinium (Gd–MSN). The ability of Gd–MSN to inhibit tumor growth was demonstrated by evaluating effects of irradiating tumor spheroids with a precisely tuned monochromatic X-ray.

## 1. Introduction

The nanotechnology that was initiated in 1960s has generated a variety of nanomaterials valuable for biomedical applications such as cancer therapy [1]. Of particular interest are nanoparticles—small particles of the size ranging from 40 to 400 nm [2,3,4]. Various materials have been used to generate nanoparticles including organic nanoparticles such as liposomes, synthetic polymers, micelles, protein and biomolecules as well as inorganic nanoparticles such as mesoporous silica nanoparticles, gold nanoparticles and diamond nanoparticles [2,3,4]. Many nanoparticles are efficiently taken up into cancer cells by endocytosis and can accumulate in late endosomes and lysosomes that are localized at the perinuclear region of a cell [5]. In addition, nanoparticles can accumulate in the tumor either via passive enhanced permeability or retention (EPR) mechanism as well as by active targeting mechanisms [2,6].

Convergence of the study on nanotechnology and the study on radiation therapy has resulted in various new approaches to enhance radiation therapy [7,8]. Radiosensitizers have been delivered to the tumor by using nanoparticles. In addition, nanoparticles containing high-Z elements have been developed. Advantageous properties of nanoparticles such as tumor targeting have important implication for radiation therapy. These nanoparticles have been evaluated in tissue culture models, in animal models and in clinical trials. A new addition to these nanoparticles is a type of mesoporous silica nanoparticles that are surface attached with gadolinium [9].

In this review, we first discuss monochromatic X-rays and the Auger effect [10] that was proposed in 1923. Various nanoparticles loaded with high-Z elements have been developed over the years [11]. Among these, we will focus on gadolinium-loaded nanoparticles, as they provide valuable reagents as a radiation sensitizer, magnetic resonance imaging (MRI) enhancing agents and as a reagent for neutron therapy [12]. We will then describe a recent study that utilized gadolinium-loaded mesoporous silica nanoparticles [9] and discuss potential significance of this study. Finally, nanoparticles developed for irradiations other than X-rays will be discussed.

## 2. Monochromatic X-ray and the Auger Effect

X-rays are electromagnetic waves in the wave length range of one picometer to ten nanometers [13]. Current radiation therapy uses white X-rays which are mixtures of these X-ray waves. By using a monochromator, the white X-ray can be separated into monochromatic X-rays each having a single energy level [14,15] and they can be used for radiation therapy. In addition, monochromatic X-rays have been used for imaging studies [16,17] to obtain sharper images. Currently, the best source for monochromatic X-ray is to use synchrotrons that generate intense X-ray beams. A synchrotron accelerates electrons to extremely high energy and then direct them to change directions by the use of magnets (bending magnets). The X-ray beams emitted from a bending magnet are directed to beamlines that are placed surrounding the synchrotron ring.

Irradiation of high-Z element such as gold, silver or gadolinium with a monochromatic X-ray causes photoelectric effect involving inner shell ionization. Destabilization of an atom is corrected by the release of photons and electrons. Figure 1 depicts one outcome that involves the release of the Auger electrons reported by Pierre Auger [10]. In this scenario, when gadolinium is irradiated with the monochromatic X-ray having an energy at or higher than the K-edge energy of gadolinium, an electron in the K-shell will be kicked out of the atom. This results in the movement of an electron from outer shell to the K-shell. This releases energy which is then used to kick out other electrons from the atom. Thus, the series of events lead to the release of electrons. If an element other than gadolinium is used, then the monochromatic X-ray energy needs to be adjusted so that it is higher than the K-edge energy of the particular element. The electrons released are called the Auger electrons that possess strong cell killing effect that involves DNA strand breaks by direct effect as well as by indirect effect mediated by radicals. In addition, cell membrane damages may be induced by the Auger electrons. In addition, a bystander effect on non-exposed cells could occur. However, due to short mean free path, the effect is relatively confined within the cancer cell. Occurrence of these electrons was reported in 1922 by Lise Meitner (discussed in [18]).

Thus, the effect of monochromatic X-rays can be amplified by the use of high-Z elements, raising the possibility that the Auger effect-based cancer therapy can be developed. There has been a quest to develop the Auger therapy (reviewed in [18,19,20,21,22,23]). In early studies, biologic effect of ^125^I-nucleotides incorporated into DNA was examined, as they undergo natural decay resulting in the release of Auger electrons. The study uncovered cell killing effect including DNA cleavage. Subsequently, photon-activation therapy (PAT) was examined. Various studies using cancer cells as well as animal tumor models have been carried out [18,19,20,21].

## 3. Various Nanoparticles Have Been Developed as Sensitizing Agents for Radiation Therapy; Focusing on Gadolinium Nanoparticles

High Z element (Au, Gd, Au, Bi etc.) containing nanoparticles have been developed as sensitizing agents for radiation therapy over the years [11]. Gold nanoparticles have been extensively studied due to their low toxicity and high electronic density which contributes to favorable amplification of radiation effects. Gadolinium-loaded nanoparticles are also of interest, as they can be used as radiation sensitizers and MRI-enhancing agents, as well as neutron capture agent [12]. Table 1 summarizes some representative nanoparticles of this type. One type of gadolinium-loaded nanoparticles uses gadolinium oxide. HA–Gd_2_O_3_ were synthesized by using a one-pot hydrothermal approach after mixing hyaluronic acid (HA) and GdCl_3_ resulting in the preparation of nanoparticles with 105-nm-diameter [24]. Radiosensitizing effect of this agent was examined using HepG2 cells as well as tumor bearing mice. Up to 9 Gy dose of X-ray was used for the experiment. GONs were synthesized using gadolinium (III) nitrate hexahydrate [25] resulting in the preparation of ultra-small nanoparticles with average diameter of 3.1 nm (hydrodynamic diameter is 8.7 nm due to hydration corona). Radiosensitizing effects of GONs were examined by irradiating (2.0 Gy/min) a variety of non-small cell lung cancer cells. Production of reactive oxygen species (ROS) and autophagy induction were detected. Gd_2_O_3_@SiO_2_ represents gadolinium oxide nanoparticles embedded in a polysiloxane shell [26]. The average diameter of this nanoparticle is 42 nm. Enhanced production of reactive oxygen species (by a factor of 1.83) was observed with the nanoparticles compared with the gadolinium chelate molecule after irradiation of mouse colon carcinoma cells CT26 with 3 or 10 Gy X-rays or with 50-keV monochromatic X-ray [26].

Ultra-small gadolinium-based nanoparticles AGuIX have been shown to be a promising type of MRI-guided radiotherapy agent [27,28,29,30,31,32]. These consist of a polysiloxane network surrounded by a number of gadolinium chelates. The size of AGuIX is small with 3 to 5-nm diameter. Verry et al. [32] reported that AGuIX nanoparticles with a diameter of 3 ± 1.5 nm cause increase in irradiation effect by a factor of 1.1–2.5 depending on the energy of photon beam used and the cell line studied. Lysosomal localization of AGuIX was demonstrated by using various human cancer cells [28]. Effective radiosensitizing effect was observed in the kiloelectronvolt region (220 kVp) as well as in the megaelectronvolt (6 MV) region in various in vitro experiments using various cancer cell lines including pancreatic cancer, glioblastoma cells, head and neck squamous cell carcinoma, cervical cell carcinoma and prostate cancer cells. Important results showing accumulation of AGuIX in the tumor and elimination by the renal route were obtained in animal model systems. Radiosensitizing effect in vivo was obtained using a wide range of animal models including mouse melanoma brain metastasis model, rat glioblastoma model and orthotopic mouse models of non-small cell lung carcinoma, mouse models of head and neck cancer, mouse model of liver cancer and rat model of chondrosarcoma [27]. AGuIX nanoparticles have recently been approved for clinical trials for multiple brain metastasis [31]. SiBiGdNP are silica-based bismuth–gadolinium nanoparticles that have a hydrodynamic diameter of 4.5 nm [33]. In vivo magnetic resonance (MR), computed tomography (CT) contrast enhancement as well as tumor suppression by 6MV clinical radiation therapy using tumor-bearing mice were observed [32].

GdW_10_@CS nanospheres are gadolinium-containing polyoxometalates-conjugated chitosan and have diameter of about 30 nm [34]. These nanoparticles were further complexed with HIF1 (hypoxia-inducible factor-1a) siRNA to suppress DNA repair. Radiosensitizing effect was shown in vitro by using hepatocellular BEL-7402 cells and HeLa cells. ROS production and DNA damage were detected. Radiosensitization in vivo was demonstrated by irradiating tumor-bearing mice with 10Gy X-ray after injection of GdW_10_@CS_siRNA_.

Albumin-based nanoparticles loaded with gadolinium have been prepared. GGD–BSA [35] and Gd–DTPA–HAS [36] are examples that were used as MRI contrast agents using tissue culture cells as well as mouse models. Gd_2_O_3_@albumin nanoparticles with average diameter of 23 nm were synthesized [37]. Photocytotoxic effects were examined using cancer cells as well as using tumor bearing mice.

A new addition to this list of gadolinium containing nanoparticles is Gd–MSN that is based on mesoporous silica nanoparticles [9]. The size of Gd–MSN used was 139-nm diameter. The radiosensitizing effect was examined using tumor spheroids as described below.

Safety of gadolinium-loaded nanoparticles is an important issue, as gadolinium-based agents are potentially nephrotoxic [38]. In the case of ultrasmall AGuIX (sub-5 nm) nanoparticles, extensive studies have been carried out in rodents as well as in nonhuman primates and the results demonstrated suitable tolerance and renal elimination of these agents [39]. The toxicity depends on the amount of agents used and the route of administration. Therefore, detailed evaluation of the safety of agents needs to be carried out for each agent.

## 4. Mesoporous Silica Nanoparticle with Surface Gadolinium Attachment Represents a Recent Addition to an Expanding List of Nanoparticles

In the next couple of sections, we describe Gd–MSN in more detail. These nanoparticles are based on mesoporous silica nanoparticles (MSN) that have a number of advantageous features among various nanoparticles including the ease of synthesis and chemical modifications [40,41,42,43,44]. They are synthesized by the sol–gel method that involves condensation of TEOS (tetraethyl orthosilicate) in the presence of templating surfactant solution. Figure 2A,B shows MSN nanoparticles used for biomedical application [45]. MSN diameter can range between 50 and 400 nm. MSN shown in figure contains 1400 pores with each pore diameter in the range of 2–4 nm—giving the appearance of honeycomb or Swiss cheese.

One of the key features of MSN is the presence of a vast surface area. Because the pore interior can be considered as a surface, this type of nanoparticles has a huge surface where various compounds can be attached; it has been estimated that MSN has 100 m^2^/g of surface area [4]. The chemistry for grafting various functional groups on the surface has been established [4]. This uses triethoxysilane derivatives with a variety of functional groups including amines, phosphonates, sulfates etc. Hydrophobic membrane like feature can also be added to the surface. Thus, methods to synthesize MSN with diverse features have been established.

To load gadolinium, amine modified MSN was first prepared by using 3-aminopropyltriethoxysilane (APTES) and then incubated with gadopentetic acid to prepare Gd–MSN [9]. Successful attachment of gadolinium to MSN was confirmed by carrying out the scanning transmission electron microscopy-energy dispersive X-ray (STEM-EDX) analysis. Along with the signal for Si and O that form Si–O–Si framework of the nanoparticle, gadolinium signal was detected (Figure 2C). The amount of gadolinium loaded onto the nanoparticle was 0.08 mg per 1 mg of MSN. Once bound, gadolinium was stably associated with MSN even after exposure to low pH or after sonication. It should be noted that the surface of Gd–MSN also contains phosphonate. This is because phosphonate surface modification provides negative charge to the nanoparticle surface and this contributes to excellent dispersibility in a solution. Therefore, two different surface modifications are necessary and the method for achieving this is described in Matsumoto et al. [9].

Tumor spheroids [46], that are three-dimensional structure of cancer cells prepared by using a special plate for growing cancer cells, provide a convenient and versatile model to evaluate gadolinium-loaded nanoparticles. Matsumoto et al. [9] utilized this tumor spheroid model to evaluate effect of monochromatic X-ray irradiation. They used tumor spheroids prepared by growing human ovarian cancer cells (The size of the spheroid is usually 0.3 mm × 0.3 mm). Incubation of the tumor spheroids with Gd–MSN overnight resulted in uniform distribution of Gd–MSN throughout the spheroid (Figure 3) as examined by carrying out confocal microscopic analysis (Figure 3B). In this experiment, ovarian cancer OVCAR8 cells expressing GFP were used so that the tumor spheroid shows green fluorescence. Gd–MSN was labeled with red fluorescent rhodamine B. Overlap of green and red fluorescence was confirmed at each focal plane of the tumor spheroid.

## 5. Monochromatic X-ray Exposure to Tumor Spheroids Incubated with Gadolinium-Loaded MSN

In the study by Matsumoto et al. [9], monochromatic X-rays generated at SPring-8 synchrotron facility located in Harima, Japan was used to irradiate the tumor spheroids that have been incubated with Gd–MSN (Figure 4A). This radiation facility uses the storage ring that is operated with a constant storage ring current of 100 mA [9]. The monochromatic X-rays of 50.25 keV from a bending magnet were shaped by horizontal and vertical slits. The incident beam size is 1.4 mm in height and 1.4-mm in width at the sample position [9]. The setup enabled generation of monochromatic X-rays with sharp band width (Figure 4B shows 50.25-keV X-ray). This X-ray beam was guided to the tumor spheroid sample and the irradiation continued for up to 1 hour. The photon flux at the sample position was 3.11 × 10^6^ photons/sec. After the exposure, the spheroids were incubated for two to three days, as cellular effect required extended time to become apparent. The results showed that the exposure of the tumor spheroids resulted in almost complete destruction of the spheroids. Spheroid destruction was dependent on the exposure time, but significant destruction was observed even after 10 minutes. The tumor spheroid destruction was correlated with the amount of Gd–MSN in the spheroid and little destruction was observed with spheroids incubated with empty MSN, suggesting that the irradiation itself is not cytotoxic under the condition used.

An important observation is that the destruction of tumor spheroid was sharply dependent on the energy level of the monochromatic X-ray used [9]. Irradiation of spheroids incubated with Gd–MSN with 50.25-keV monochromatic X-ray (just above the K-edge energy of gadolinium) almost completely destructed the spheroids, while the irradiation with 50.0-keV monochromatic X-ray did not cause effect (Figure 4C). Irradiation with 50.4-keV X-ray destructed the spheroids, but the extent was less than that observed with the 50.25-keV X-ray. Thus, a dramatic difference in the tumor destruction effect can be observed by slightly changing the energy of monochromatic X-ray. It is also interesting that the X-ray irradiation had little effect in the absence of gadolinium. These results suggest that the use of a monochromatic X-ray with a defined energy (50.25 keV in this case) in combination with gadolinium-loaded nanoparticles is effective in tumor destruction. Recently, a compact laser-driven synchrotron X-ray source has been developed by Eggl et al. [47], raising the possibility that monochromatic X-rays will be used widely in clinical settings in the future.

## 6. Further Potential of Using Gd–MSN

For possible clinical translation of Gd–MSN, it is important to address the issue of safety. While MSN nanoparticles are biocompatible and safe [48], they are slowly degraded and may cause retention in the body. Therefore, efforts have been made recently to confer biodegradability to MSN so that their degradation is enhanced [49,50,51,52,53,54]. This involved incorporating biodegradable bonds into the framework of the nanoparticle by employing the chemistry that led to the development of periodic mesoporous silica nanoparticles (PMO). These nanoparticles are called biodegradable periodic mesoporous organosilica (BPMO) and the synthesis involved the use of bridged alkoxysilane precursor instead of TEOS used for the synthesis of MSN [49,54]. The bridged alkoxysilane is made up of two silane units connected by a chemical bond. The biodegradable chemical bonds that have been used include di- and tetrasulfide bonds that are cleavable by reducing conditions such as those encountered inside the cell [49]. In addition, protease sensitive bonds have been used [54]. Degradation in vitro of BPMO and delivery of anticancer drugs by BPMO has been reported [49,54]. In a study using BPMO with tetrasulfide bonds [54], it was found that BPMO is degraded after incubation with glutathione for three days, while MSN remains intact even after incubation for seven days. It is expected that gadolinium can be loaded onto BPMO by using the method that was used for MSN.

## 7. Tumor Organoids as a Convenient Tumor Model to Characterize Nanomaterials Loaded with High Z Element

In the experiment described above, tumor spheroids were used to examine the efficacy of gadolinium loaded MSN. Tumor spheroids are three-dimensional arrangement of cancer cells that resemble tumor mass. They are formed by using a special plate that prevents attachment of cells to plate surface [46]. It is believed that the three-dimensional tumor represents human tumor better than two-dimensional tissue culture cells. The center of tumor spheroids often contains a necrotic area that again resembles human tumor. Response to anticancer drugs is different with tumor spheroids compared with tissue culture cells. While tumor spheroids consist only of cancer cells, other types of cells such as stromal cells and macrophages can be added to mimic tumor microenvironment [55,56]. Furthermore, recent efforts involve using patient tumor derived cells to develop tumor organoids. These models are expected to closely mimic patient tumors and the study using these may contribute to the development of precision medicine that has been proposed as a new direction in medical therapy [57,58].

## 8. Irradiation Other than X-Rays and Development of Various Nanomaterials

Instead of X-rays, other types of irradiation are used for medical therapy. For example, boron neutron capture therapy (BNCT) involves exposure of boron-10 (^10^B) to thermal neutron resulting in the splitting of boron atom to lithium and helium [59,60,61]. Helium nucleus is an α-particle that has strong destructive effect including DNA damage. Clinical studies were initiated right after world war II and then more extensively in 1980s and 1990s [59,60]. Boron compounds used for BNCT include BPA (boronophenylalanine) and BSH (sodium borocaptate). In clinical settings, these reagents are administered by intravenous injection into patients. Nanoformulated boron reagents such as boron-loaded liposomes, polymers and silica nanoparticles have been developed [62,63,64,65].

Exposure of gadolinium to thermal neutron releases γ-ray as well as electrons leading to destruction of cancer cells. This method is called GNCT (gadolinium neutron capture therapy) and has been evaluated as an alternative to BNCT [66]. The neutron capture cross section of ^157^Gd is much higher than that of ^10^B and that its natural abundance is high. A difference between BNCT and GNCT is that α-ray is emitted from boron-10 in the case of BNCT, while γ-ray is emitted from gadolinium in the case of GNCT. Because α-ray’s mean free path is more limited than γ-ray, effect on tumor is expected to be more focused with BNCT. Further investigation is needed to compare BNCT and GNCT.

Proton beams are widely used as an effective cancer therapy, as proton beams can be optimized so that the energy level at the site of tumor will be high, in contrast to X-ray that loses energy as it penetrates into the tissue. There are many proton therapy centers that are carrying out cancer therapy. This raises the possibility that the proton beam can be used to combine with various elements delivered to the tumor by the use of nanomaterials. The first demonstration of this possibility was reported by Cirrone et al. [67]. In this work, proton beam was irradiated on boron-11 in BSH (sodium borocaptate) resulting in the release of α-rays (proton exposure of boron-11 results in the release of three α-particles). Cell death and DNA damage were observed with breast cancer cells. This therapy is called proton boron capture therapy (PBCT) and may turn out to be a powerful alternative to BNCT. Another irradiation method uses carbon ion beams that have been used to ionize molecules inside cancer cells [68]. Effects on cancer cells are reported to be substantial and DNA damage appears to occur within a narrow region of DNA. It has been reported that gadolinium-based nanoparticles AGuIX enhances effect of carbon ion irradiation in human tumor cells [69]. Advance in the study of radiation methods and the development of various nanomaterials could greatly enhance the advance in radiation therapy.

## 9. Summary

Monochromatic X-rays provide a valuable source for radiation therapy. This special type of X-ray with a single energy level can be obtained by separating a synchrotron-generated white X-ray by the use of a monochromator. Irradiation of high-Z elements such as gadolinium causes photoelectric effect including the release of Auger electrons. Various nanoparticles containing gadolinium have been developed for their effect to enhance radiation therapy and a recent addition to the growing list of such nanoparticles is gadolinium-loaded mesoporous silica nanoparticles (Gd–MSNs). These nanoparticles differ in their size and composition. Additional nanoparticles may be developed in the future. Thus, it will be important to evaluate their ability to enhance effect of X-ray irradiation. While cancer cell models and animal models will continue to be important, tumor organoid models provide versatile and convenient assays to examine their ability to cause destruction upon X-ray irradiation.

## Figures and Tables

**Figure 1 nanomaterials-10-01341-f001:**
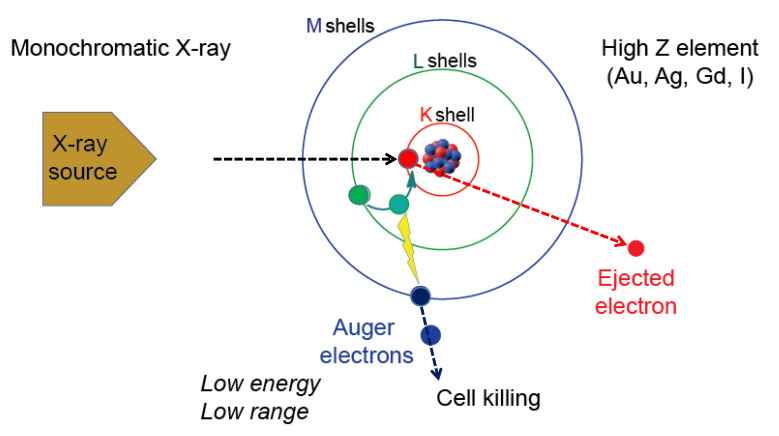
The Auger effect. Irradiation of high-Z element such as Au, Ag, Gd or I with a monochromatic X-ray having a defined energy level causes effects that include ejection of a K-shell electron (red circle). One of the ways to reestablish energy equilibrium is the movement of an electron from a higher shell (green circle) causing the release of energy that is used by another electron (blue circle) which is released as an Auger electron. The Auger electron has strong cell killing effect.

**Figure 2 nanomaterials-10-01341-f002:**
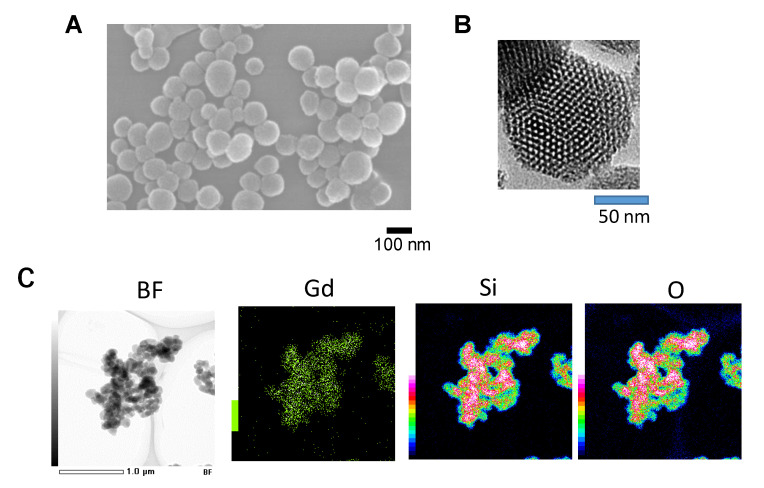
Mesoporous silica nanoparticles (MSN) prepared by the sol–gel method. (**A**) Scanning electron microscope (SEM) picture; (**B**) Transmission electron microscope (TEM) picture; (**C**) Scanning transmission electron microscopy-energy dispersive X-ray (STEM-EDX) images of Gd–MSN. Bright field image as well as elemental mapping images of Gd, Si and O are shown. Modified from [9].

**Figure 3 nanomaterials-10-01341-f003:**
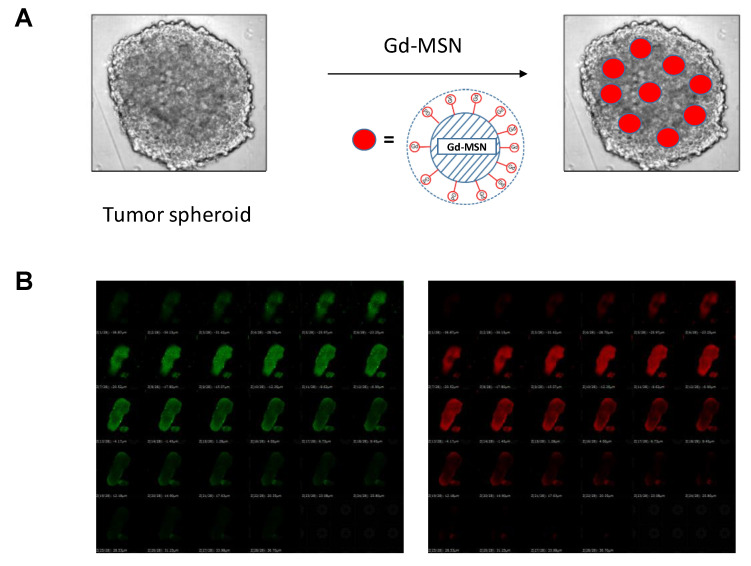
(**A**) Schematic drawing showing tumor spheroids incubated with Gd–MSN results in the distribution of Gd–MSN throughout the spheroid. Position of Gd–MSN in the spheroid is arbitrary to emphasize the presence of Gd–MSN in the spheroid; (**B**) tumor spheroid prepared from GFP-expressing cancer cells is observed as a green mass. When the spheroid was incubated with gadolinium-loaded MSN labeled with rhodamine B, uniform distribution of the nanoparticle throughout the spheroid was observed by using confocal microscopy (see the red fluorescence). Focal planes from the top to the bottom of the spheroid sample is shown. In all planes, green fluorescence of the cancer cells overlaps with the red fluorescence of Gd–MSN. Modified from [9].

**Figure 4 nanomaterials-10-01341-f004:**
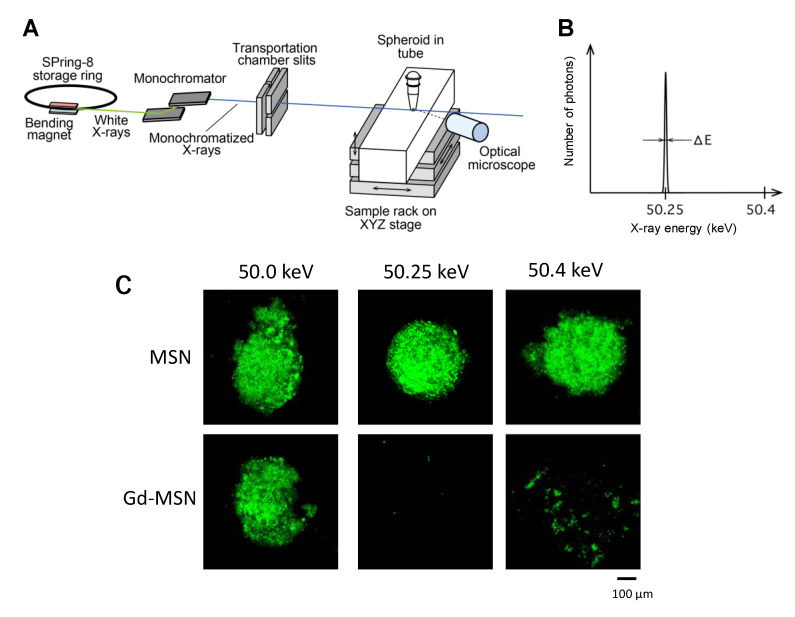
Irradiation with monochromatic X-ray. (**A**) Experimental setup; (**B**) band width of the 50.25-keV monochromatic X-ray; (**C**) tumor spheroids incubated with Gd–MSN were irradiated with 50.0, 50.25- or 50.4-keV monochromatic X-ray for 20 min and then incubated for three days. Tumor spheroids with Gd–MSN after irradiation with 50.25 or 50.4 keV were destructed, while spheroids with empty MSN were not affected by the irradiation. Modified from [9]

**Table 1 nanomaterials-10-01341-t001:** Various nanoparticles loaded with gadolinium. Some representative nanoparticles used as radiosensitizing agents as well as magnetic resonance imaging (MRI) enhancing agents are shown. The dose of X-ray and the animal models used are described in the text.

Radiosensitization Effect					
**NPs**	Size(nm)	In Vitro	In Vivo	Biological Effect	Reference
**Gadolinium oxide NPs**					
**HA-Gd_2_O_3_**	105	Hep G2	Mouse xenograft	-	[24]
**GONs**	3.1	NSCLC	-	ROS autophagy	[25]
**Gd_2_O_3_@SiO_2_**	42	CT26	-	ROS	[26]
**Polysiloxane-Gd chelates**					
**AGulX**	3	Panc-1, SQ20B, B16F10, U87MG, HeLa	Brain metastasis	ROS	[27,28,29,30,31,32]
			Rat brain tumor	DNA damage	
**SiBiGdNP**	4.5	A549 NSCLC	Mouse xenograft	DNA damage	[33]
**Polyoxometrates-conjugated chiotosan**					
**GdW10@CS**	30	BEL-7402	Mouse xenograft	ROS	[34]
		HeLa		DNA damage	
**Albumin NP-s-GD-DTPA**					
**Gd_2_O_3_@BSA**	23.3	HepG2	-	ROS	[35]
		RAW264.7		photocytotoxicity	
**MSN loaded with Gd**					
**Gd-MSN**	139	OVCAR8	-	Tumor spheroid destruction	[9]

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
