# Peer review of "Studies on the Exposure of Gadolinium Containing Nanoparticles with Monochromatic X-rays Drive Advances in Radiation Therapy"

_nanomaterials, 2020, doi:10.3390/nano10071341_

Round 1

Reviewer 1 Report

General comments

This is an interesting review, which first discuss the use of monochromatic X-rays in combinations with various nanoparticles (NPs), the Auger effect, and the use of different NPs loaded with high Z elements. Then a recent study that utilized gadolinium-loaded mesoporous silica NPs is described. Finally, NPs developed for irradiations of other than X-rays is presented.

The ideal requirements for designing an effective NP as a diagnostic and therapeutic tool are: controlled particle size, surface modification, functionalization, binding behavior, solubility to aqueous environment, low toxicity, biocompatibility, targeting, permeability, retention and biodegradation. The review does need to address all these questions, but to make the review somewhat more complete, I would propose that at least the importance of size and shape of the NPs, as well as some comments on the importance of the chemical composition and surface treatment are mentioned. It is well known that both the maximum cell uptake by NPs (which has been suggested to be between 20 and 60 nm) and the biodistribution and route of elimination from the body are strongly depending on the size of the NPs. To avoid accumulation of NPs in organs such as heart and liver, causing potential long-term side effects, metal NPs should be eliminated from the body within a few days.

Specific comments

The title ”Convergence of the study on monochromatic X-rays and the research on nanoparticles drives advance in radiation therapy” does not really reflect the content in the ms. I would suggest something like “Exposure of nanoparticles with monochromatic X-rays drives advances in radiation therapy”.

Page 3. Figure 1: On the figure it is written “Cell killing, DNA damage, apoptosis”. However, Auger electrons (AEs) may cause DNA double-strand breaks by both direct and indirect effects mediated by  radicals due to interaction with water molecules. AEs may also cause cell membrane damage leading to cell death. There is a localized short-range “cross-dose” effect of AEs on cancer cells which are directly adjacent to targeted cells, and a longer range “bystander” effect on more distant cells. Unrepaired DNA damage can then result in cell death by several pathways, e.g. apoptosis or mitotic catastrophe. The damage and cell killing caused by the AEs shluld therefore be explained in more details.

Page 4, line 122: It is written that the size of AGuIX is small with 5 nm diameter. However, even smaller sizes have been studied. In the ref. 31 in the ms (Bort, G.; Lux, F.; Dufort, S.; Cremillieux, Y.; Verry, C.; Tillement, O. EPR-mediated tumor targeting using ultrasmall-hybrid nanoparticles: From animal to human with theranostic AGuIX nanoparticles. Theranostics 2020, 10, 1319-1331.), the use of AGuIX with a diameter of 4 ± 2 nm is described. In e.g. Verry C, Sancey L, Dufort S, et al. Treatment of  multiple brain metastases using gadolinium nanoparticles and radiotherapy: NANORAD, a phase I study protocol.  BMJ Open 2019;9:e023591. doi:10.1136/bmjopen-2018-023591, the authors present a study using AGuIX with a diameter of 3±1.5 nm. In the same paper it is written that addition of AGuIX (from 0.1 mM to 1.0 mM) increases the irradiation effectiveness by a factor of 1.1–2.5 depending on the energy of the photon beam used (from 6 MV to 50 kV, respectively) and the cell line studied. This reference, with some comments about the results presented, could be added to the ms.

Page 9, chapter “6. Irradiation other than X-rays and development of various nanomaterials”: When GNCT (gadolinium neutron capture therapy) is discussed it might be worth mentioned that the neutron capture cross sections of 155Gd and 157Gd are very high and that 155Gd and 157Gd are present at relatively high yields naturally (14.8 and 15.65% natural abundance, respectively).

It would be good to have some kind of summary and/or conclusion, in the end of the ms. In this/these chapters the results in the paper should be summarized and concluded.

Author Response

Comment 1: The title ”Convergence of the study on monochromatic X-rays and the research on nanoparticles drives advance in radiation therapy” does not really reflect the content in the ms. I would suggest something like “Exposure of nanoparticles with monochromatic X-rays drives advances in radiation therapy”.

Answer: Thank you very much for the suggestion. We changed the title to “Studies on the exposure of gadolinium containing nanoparticles with monochromatic X-rays drive advances in radiation therapy”.

Comment 2: Page 3. Figure 1: On the figure it is written “Cell killing, DNA damage, apoptosis”. However, Auger electrons (AEs) may cause DNA double-strand breaks by both direct and indirect effects mediated by radicals due to interaction with water molecules. AEs may also cause cell membrane damage leading to cell death. There is a localized short-range “cross-dose” effect of AEs on cancer cells which are directly adjacent to targeted cells, and a longer range “bystander” effect on more distant cells. Unrepaired DNA damage can then result in cell death by several pathways, e.g. apoptosis or mitotic catastrophe. The damage and cell killing caused by the AEs should therefore be explained in more details.

Answer: Thank you for the comment about cellular effect of Auger electrons. We added the following sentences to the text (revised ms page 2, lines 77-80). “The electrons released are called the Auger electrons that possess strong cell killing effect that involves DNA strand breaks by direct effect as well as by indirect effect mediated by radicals. Also, cell membrane damages may be induced by the Auger electrons. In addition, a bystander effect on non-exposed cells could occur.”.

Comment 3: Page 4, line 122: It is written that the size of AGuIX is small with 5 nm diameter. However, even smaller sizes have been studied. In the ref. 31 in the ms (Bort, G.; Lux, F.; Dufort, S.; Cremillieux, Y.; Verry, C.; Tillement, O. EPR-mediated tumor targeting using ultrasmall-hybrid nanoparticles: From animal to human with theranostic AGuIX nanoparticles. Theranostics 2020, 10, 1319-1331.), the use of AGuIX with a diameter of 4 ± 2 nm is described. In e.g. Verry C, Sancey L, Dufort S, et al. Treatment of multiple brain metastases using gadolinium nanoparticles and radiotherapy: NANORAD, a phase I study protocol.  BMJ Open 2019;9:e023591. doi:10.1136/bmjopen-2018-023591, the authors present a study using AGuIX with a diameter of 3±1.5 nm. In the same paper it is written that addition of AGuIX (from 0.1 mM to 1.0 mM) increases the irradiation effectiveness by a factor of 1.1–2.5 depending on the energy of the photon beam used (from 6 MV to 50 kV, respectively) and the cell line studied. This reference, with some comments about the results presented, could be added to the ms.

Answer: We added the following sentences to describe the use of AGuIX nanoparticles with a diameter less than 5 nm (revised ms, page 4, lines 127-129). “The size of AGuIX is small with 3 to 5 nm diameter. Verry et al reported that AGuIX nanoparticles with a diameter of 3±1.5 nm cause increase in irradiation effect by a factor of 1.1-2.5 depending on the energy of photon beam used and the cell line studied.”

Comment 4: Page 9, chapter “6. Irradiation other than X-rays and development of various nanomaterials”: When GNCT (gadolinium neutron capture therapy) is discussed it might be worth mentioned that the neutron capture cross sections of 155Gd and 157Gd are very high and that 155Gd and 157Gd are present at relatively high yields naturally (14.8 and 15.65% natural abundance, respectively).

Answer: We added the following sentence to mention this point. (revised ms, page 10, lines 303-304). “The neutron capture cross sections of 157Gd is much higher than that of 10B and that its natural abundance is high”.

Comment 5: It would be good to have some kind of summary and/or conclusion, in the end of the ms. In this/these chapters the results in the paper should be summarized and concluded.

Answer: We added a summary at the end of the manuscript.

Reviewer 2 Report

The Authors have submitted a review mostly regarding the potential employment of silica-Gd nanoparticles as radiotherapy sensitizers.

The idea behind the review can be acceptable. On the other hand, this manuscript cannot be considered for publication at this stage for the following points:

-The presentation has to be completely re-structured regarding the English style. Moreover, there are several "subject jumps" without a direct linkage.

-This manuscript has been submitted as a review. On the other hand, it seems composed by mixing some research papers with a quite extended intro. For example, lines 154-205 seems copy/paste from research papers and not a discussion on past research.

-Authors report a short discussion about silica degradation. They should restructure that section, as silica can be degradable as reported in many papers, for example doi: 10.1016/j.nano.2018.05.007, 10.1002/adma.201604634, and 10.1002/adtp.202000022.

Author Response

Comment 1: The presentation has to be completely re-structured regarding the English style. Moreover, there are several "subject jumps" without a direct linkage.

Answer: The impression about “subject jumps” is perhaps caused by abrupt introduction of different subject matter. To correct this point, we added sentences to provide smooth transition of a subject matter (revised ms, page 5, lines 168-170; page 8, lines 252-254).

Comment 2: This manuscript has been submitted as a review. On the other hand, it seems composed by mixing some research papers with a quite extended intro. For example, lines 154-205 seems copy/paste from research papers and not a discussion on past research.

Answer: Every sentence I write is written completely new. If they look similar to our previous publications, that is because we discuss these subject matters multiple times. In all my 182 publications, I have never copy/pasted any sentence. I would also like to mention that I have been serving as a series editor for The Enzymes published by Academic Press/Elsevier since 2000 and I am keenly aware of the problem of copy/paste.

Comment 3: Authors report a short discussion about silica degradation. They should restructure that section, as silica can be degradable as reported in many papers, for example doi: 10.1016/j.nano.2018.05.007, 10.1002/adma.201604634, and 10.1002/adtp.202000022.

Answer: While mesoporous silica nanoparticles can be degraded, this depends on incubation conditions and the extent of degradation is variable. Introduction of biodegradable bonds into the framework will confer consistent degradability. We have revised the first two sentences in this section to describe this point. “While MSN nanoparticles are biocompatible and safe, they are slowly degraded and may cause retention in the body. Therefore, effort has been made recently to confer biodegradability to MSN so that their degradation is enhanced”. (revised ms, page 9, lines 260-262)

Reviewer 3 Report

In this manuscript, the authors review the recent development in the nanoparticle-based radiation therapy with a special emphasis on the nanoparticles containing gadolinium using monochromatic x-ray radiation. The authors give a brief introduction of the history and the mechanism of radiation therapy, which is very useful for the material scientists looking for biomedical applications of their novel nanomaterials. The authors also list several key experiments for the applications of nanoparticles in radiation therapy both in vivo and in vitro. The advantages of a monochromatic x-ray for radiation therapy are also discussed. Furthermore, the authors explain the use of organoids as 3D model system for examining the effect of monochromatic x-ray radiation. In the end, the authors discuss the future direction and the potential application of gadolinium nanoparticles for radiation therapy other than x-ray. In general, we believe that this review is very useful for scientists interested in the biomedical applications of nanomaterials. To help the reads to better understand the development in this field. We would like to suggest that the authors to add more discussion on the following issues:

  1. It will be helpful to add a table to summarize the applications of the nanoparticle-based radiation therapy for different cancer models.
  2. The monochromatic x-ray source discussed in this paper is generated from synchrotron radiation. Is there another source that can produce a monochromatic x-ray for radiation therapy?
  3. It is a good idea to compare the radiation dosage for current clinical radiation therapy and the nanoparticle-based radiation therapy.
  4. The advantages of gadolinium nanoparticle-based radiation therapy over the other nanoparticles such as gold should be discussed more in detail.
  5. The toxicity and biodistribution of gadolinium nanoparticles should be discussed.

Author Response

Comment 1: It will be helpful to add a table to summarize the applications of the nanoparticle-based radiation therapy for different cancer models.

Answer: Description of different cancer models used in the experiments in Table 1 is included in the text. To further increase this information, we revised a sentence in the text to the following. “Radiosensitizing effect in vivo was demonstrated using a wide range of animal models including mouse melanoma brain metastasis model, rat glioblastoma model and orthotopic mouse models of non-small cell lung carcinoma, mouse models of head and neck cancer, mouse model of liver cancer and rat model of chondrosarcoma”. (revised ms, page 4-5, lines 135-139).

Comment 2: The monochromatic x-ray source discussed in this paper is generated from synchrotron radiation. Is there another source that can produce a monochromatic x-ray for radiation therapy?

Answer: Yes, a compact laser-driven synchrotron source for monochromatic X-rays has been reported by Eggl et al. This information is added to the revised manuscript. The following sentence is added. “Recently, a compact laser-driven synchrotron X-ray source has been developed by Eggl et al”.(revised ms, page 8-9, lines 254-256)

Comment 3: It is a good idea to compare the radiation dosage for current clinical radiation therapy and the nanoparticle-based radiation therapy.

Answer: The dose of X-ray used in nanoparticle-based radiation therapy experiments is described in the text in section 3. In general, 3 to 10 Gy X-rays were used. To clarify this, we added a sentence in the legend to Table 1.

Comment 4: The advantages of gadolinium nanoparticle-based radiation therapy over the other nanoparticles such as gold should be discussed more in detail.

Answer: Thank you for mentioning about gold. We added the following sentence to mention about the gold nanoparticles at the beginning of section 3 before discussing gadolinium nanoparticles. “Gold nanoparticles have been extensively studied due to their low toxicity, good biodistribution and their high electronic density which contributes to favorable amplification of radiation effects.” (revised ms, page 3, lines 102-104).

Comment 5: The toxicity and biodistribution of gadolinium nanoparticles should be discussed.

Answer: Thank you for pointing this out. We added a paragraph at the end of section 3 to address this safety issue. (revised ms, page 5, lines 158-163)

Round 2

Reviewer 2 Report

I regret the Authors have intended my point as "plagiarism". I have never meant that and I did not detect any kind of plagiarism.

My point was (and is) related to the presentation of the review. In particular in section 4-5-6-7, some points are more a description typical of a research article (just and example, rows 202-210, 222-224, 255-269, 309-317) than an authoritative illustration of the field.

Moreover, it is puzzling the introduction of "tumor spheroids" from row 225 or of "precision medicine" in row 319 without direct linkages with the former paragraphs.  

Overall, I am very sorry but I can not support this manuscript for publication at this time. It may became publishable only after a complete re-styling.

Author Response

I regret the Authors have intended my point as "plagiarism". I have never meant that and I did not detect any kind of plagiarism.

Answer: Thank you very much for mentioning that the reviewer has not detected any kind of plagiarism.

My point was (and is) related to the presentation of the review. In particular in section 4-5-6-7, some points are more a description typical of a research article (just and example, rows 202-210, 222-224, 255-269, 309-317) than an authoritative illustration of the field.

Answer: We have made changes to address this point. (page 6, lines 201-203; page 7, lines 225-227).

Moreover, it is puzzling the introduction of "tumor spheroids" from row 225 or of "precision medicine" in row 319 without direct linkages with the former paragraphs.  

Answer: We agree that the description on precision medicine was a bit out of context. Therefore, we have modified the sentences (page 9, lines 290-293).